# Development of an Open-Source Injection Mold Monitoring System

**DOI:** 10.3390/s23073569

**Published:** 2023-03-29

**Authors:** Tiago E. P. Gomes, Mylene S. Cadete, Jorge A. F. Ferreira, Renato Febra, João Silva, Tiago Noversa, António J. Pontes, Victor Neto

**Affiliations:** 1TEMA—Centre for Mechanical Technology and Automation, Department of Mechanical Engineering, University of Aveiro, 3810-193 Aveiro, Portugal; tiago.emanuel.gomes@ua.pt (T.E.P.G.);; 2Geco—Gabinete Técnico e Controlo de Moldes em Fabricação Lda, 2405-032 Maceira, Portugal; 3CeNTI—Centro de Nanotecnologia e Materiais Técnicos, Funcionais e Inteligentes, 4760-034 Vila Nova de Famalicão, Portugal; 4IPC—Institute for Polymers and Composites, Universidade do Minho, 4800-058 Guimarães, Portugal

**Keywords:** injection molding tool monitoring, information technology, data acquisition, graphical user interface, user-computer interface, open source software, low cost, real time

## Abstract

In the highly competitive injection molding industry, the ability to effectively collect information from various sensors installed in molds and machines is of the utmost relevance, enabling the development of data-based Industry 4.0 algorithms. In this work, an alternative to commercially available monitoring systems used in the industry was developed and tested in the scope of the TOOLING 4G project. The novelty of this system is its affordability, simplicity, real-time data acquisition and display in an intuitive Graphical User Interface (GUI), while being open-source firmware and software-based. These characteristics, and their combinations have been present in previous works, but, to the authors’ knowledge, not all of them simultaneously. The system used an Arduino microcontroller-based data acquisition module that can be connected to any computer via a USB port. Software was developed, including a GUI, prepared to receive data from both the Arduino module and a second module. In the current state of development, data corresponding to a maximum of six sensors can be visualized, at a rate of 10 Hz, and recorded for later usage. These capabilities were verified under real-world conditions for monitoring an injection mold with the objective of creating the basis of a platform to deploy predictive maintenance. Mold temperature, cavity pressure, 3-axis acceleration, and extraction force data showed the system can successfully monitor the mold and allowed the clear distinction between normal and abnormal operating patterns.

## 1. Introduction

The mold is at the heart of the injection molding process. In its cavity, molten plastic is transformed into a solid part. Thus, sensors to monitor in- and on-mold process and tool parameters have been adapted and developed, providing valuable data relating to part quality and mold condition [1]. The relevance and usefulness of such sensors is maximized in the scope of Industry 4.0, as data collection and storage capabilities improve, and data-driven prediction models become more accurate [2].

An activity that, while necessary, leads to machine down-time that is maintenance, accounting for a significant portion of operational costs [3]. Through the usage of sensors in the mold and algorithms developed from/based on the collected data, predictive maintenance becomes possible, allowing for the detection of operational patterns leading to or indicative of a given fault [4]. This, in turn leads to less rejected parts [5], and enables programmed corrective action, reducing machine down-time and inefficiency. Parameters such as cavity pressure and temperature are commonly monitored to assess or predict part quality [6], or to automate process control [7], but can also be used for predictive maintenance [4]. Other mold monitoring sensors previously suggested for predictive maintenance include strain gauge sensors [4], 3D accelerometers [8], and custom pressure sensors [8]. Furthermore, less conventional approaches have been studied, and show potential for being used for this purpose in the future, as is the case of acoustic signals collected from microphones installed near the mold and machine [9]. Data from these sensors and/or other sensors installed in the mold and machine, as well as contextual data can be collected, processed, and used to develop predictive maintenance algorithms, as seen in the work of Nunes et al. [10], through the application of generalized fault trees and abnormality detection. Furthermore, mold monitoring sensors data can also be used develop machine learning algorithms to optimize process parameters for part quality prediction and improvement [11,12].

The high quality and highly reliable data acquisition systems typically used for injection-molding monitoring, such as KISTLER’s ComoNeo [13], PRIAMUS’ FILLCONTROL or BlueLine Hardware and QFlow Systems Engineering [14], implement proprietary monitoring and control solutions to acquire data from some of the previously mentioned sensors. These systems are, however, economically demanding and closed, often posing a barrier to the development and testing of customized new hardware and software solutions. On this scope, the creation of open-source software and firmware data acquisition systems, which can be based on lower cost hardware and/or excuse the payment of expensive proprietary software license fees, is a solution that brings opportunities for a more widespread and diversified implementation of some core Industry 4.0 concepts to injection-molding mold monitoring [15]. These systems allow academic, research and development entities, as well as companies to take part in the development of new, fit for purpose solutions, dedicated to solving each specific set of problems, often not solvable with the commercially available systems and solutions. Systems that answer similar challenges are described in published literature, applied to aerodynamic studies or mechanical components wear monitoring [16,17]. In these, software developed with MATLAB, LabView or Python, among others, allows data visualization and storage.

In the injection-molding field, a low-cost monitoring system has been developed by Silva et al. [18] using software written in Python programming language and implemented on a Raspberry Pi single board computer. The system allowed data visualization at each cycle end and was used to monitor legacy injection molding machines parameters, successfully helping detect faults in real world context. A different work describes the implementation of a custom pressure sensor and a 3D accelerometer in the injection-molding tool for real-time monitoring towards smart predictive maintenance [8]. Data acquisition was performed through a commercially obtained capacitance to digital converter for the pressure sensor, and an analog-to-digital converter for the 3D accelerometer, connected to a microcontroller that sent the data to a computer through a Bluetooth module. No information is given regarding the used software by the authors in the publication. Kusić and Slapšak proposed a low cost IoT-enabled system to monitor temperature and cavity pressure [19]. Their system used Bluetooth, Wi-Fi and the MQTT protocol for communication between sensors and devices, acquired data at a 2–3 Hz rate, and featured a web-based GUI. An open-source machine and mold monitoring software based on an industrial Raspberry Pi has been prototyped and tested by Ogorodnyk et al. [20]. It allows the acquisition of data in hard real-time at a sampling rate of 2 Hz, or higher for soft real-time, registering values for up to 97 machines and process parameters.

In the injection molding industrial practice and literature, there is a scarcity of mold monitoring solutions that can meet the needs of smaller companies, research and education institutions. Such a system needs to combine affordability, so it can be attained or assembled and used on lower budgets. Furthermore, it needs to be kept as simple as possible, to allow a wider range of operators to use it and service it. Simultaneously, data needs to be collected in real-time and intuitively conveyed via a GUI, allowing for prompt action from the operator on the process according to the displayed data or messages, if needed. In addition, to maximize the innovation potential of the created system, it needs to be made as open-source as possible, allowing for incremental innovation stemming from a community of users, leading to improvement of the system itself, and for it to be modified and employed in different real-world situations with specific requirements not considered originally. As verified in published literature, there are systems each with a combination of one or more of these features, but, to the authors’ knowledge, none that meets them all. Therefore, in the context of a case study aimed at collecting data to develop predictive maintenance algorithms, this paper describes the creation of an open-source software and firmware-based low-cost monitoring system encompassing all the above-mentioned characteristics. It allowed the acquisition of data relative to mold cavity pressure, mold plate temperature, as well as extraction force, and mold vibrations in real time. Furthermore, a GUI displaying the real time data during the process was also successfully implemented. The real-world performance of our system was tested and its validity as a potential basis for later implementation of predictive maintenance algorithms was shown.

## 2. Materials and Methods

### 2.1. Hardware

The main components used to develop and implement the mold monitoring system are listed and numbered on Table 1, along with their respective designations. The assembled components, identified in accordance with Table 1 numbering can be observed in Figure 1. The sensors were selected to allow measuring multiple process variables, namely temperature, cavity pressure, force, and vibration, via an inertial module with a 3D accelerometer and a 3D gyroscope. Since the force and pressure signals obtained from the sensors were in the order of 10^−11^ C/N and C/bar, the sensors were connected to their respective charge amplifiers, which convert these signals into 0 to 10 V output voltages. The charge amplifier assigned to the force sensor, in accordance with its measuring range, is programmable through RS232. It is, therefore, possible to adjust its range to the sensor sensitivity and quantities being monitored. A circuit based on the Integrated Circuit (IC) MAX232 was created to enable the amplifier to be programmed through the Arduino microcontroller.

The electronic circuit developed for the interface between the charge amplifiers and the Arduino is represented in Figure 2. Two thermocouple signal amplifiers were also connected to the Arduino, and their pins and respective connections can also be seen in the figure. The MCP9600 thermocouple signal amplifiers were specifically chosen for their versatility, since they allow the Analog to Digital Converter (ADC) resolution definition, thus allowing the adjustment of conversion time to the required sampling rate.

### 2.2. Firmware

Arduino IDE was used to program Arduino. Three working modes were defined in the code, available online at https://github.com/TEPGomes/OpenMMS-T4G (accessed on 23 March 2023), as mentioned in the Appendix A. In measuring mode, the charge amplifiers are activated with signals sent from the Arduino’s Pulse Width Modulation (PWM) pins and data is read from each combination of sensor, amplifier, and ADC. Temperature values are directly stored in a variable, while the pressure and force values need to be calculated before being registered. These can be obtained through Equation (1),
(1)$$x=\frac{x_{max}}{V_{max}}V_{read},$$
in which x is the variable value to be calculated (force in N or pressure in bar), $x_{max}$ is the maximum value of this variable measurable within the charge amplifier range, $V_{max}$ is the maximum voltage that corresponds to $x_{max}$, in this case, the Arduino’s analog pins 5 V maximum voltage input. $V_{read}$ is the voltage read in the input analog pin. Its value can be calculated through Equation (2),
(2)$$V_{read}=\frac{V_{max}}{1023}A_{read},$$
where $A_{read}$ is the ADC output value, 1023 is the decimal number corresponding to the biggest decimal number representable by a ten-bit binary number. $x_{max}$ can be obtained through Equation (3):(3)$$x_{max}=\frac{Q_{max}}{S},$$
in which $Q_{max}$ is the charge corresponding to the maximum defined range in the charge amplifier, and S, the sensor sensitivity. The values received and converted for each sensor are concatenated into a string and sent to the computer.

In the waiting or pause mode, the Arduino stops reading data from the sensors until the next reading mode initiation message is received. The last operating mode allows communication with the force sensor charge amplifier, and in the current system version it is only accessible through a serial port monitor on operator command.

### 2.3. Software

To develop the system software, that can be viewed and downloaded from https://github.com/TEPGomes/OpenMMS-T4G (accessed on 23 March 2023), as stated in the Appendix A, Python programming language was used. Qt Designer was employed to create the GUI. It allows setting up the graphical layout of each window through a graphical interface. The library pyqt5 was used for association between actions, signals, and events with the GUI elements, as well as integration with the remaining software functionalities. In Figure 3, a GUI schematic representation can be seen, where each window’s possible actions and operation state have been highlighted. An image of the main window during a test monitoring session, with the main zones highlighted and identified, can be observed in Figure 4. Buttons to interact with the window can be found at the top zone, the middle zone is reserved for visualization graphs display, and the bottom area is used to display messages for the user with the system status and selected options.

When developing a GUI, it must be considered that users should have permanent access to the open windows’ active functions. This becomes critical when computationally intensive tasks simultaneously need to be executed. This is the case with the application described in this work, in which it is important to assure data visualization with real-time graphs, GUI responsiveness, and communication with the data acquisition modules. Parallel computing was used to allow it, through multithreading, as represented in Figure 5. Four thread types were created using pyqt5’s Qthread class, with the main one being relative to the graphical GUI. The second and third were created for connection and communication with data acquisition modules. The las thread type was developed for updating the graphs with the data being received. Communication between threads is implemented through signals leading to the execution of specific functions.

Data acquisition modules communication was implemented through the pyserial library, with Figure 6 showing a schematic representation of the code used to do so. As defined in the Firmware Subsection, once the devices are successfully connected, as soon as the start signal is received by the thread, the character for sensor reading initiation is emitted. A timer is initiated and used for registering the time when each new set of data is received. Data is received as a character comma-separated string of sensor reading values, which, after being separated and associated with their timestamp, are put into a queue, method used to transfer data to the graph-allocated threads. A signal for reading these data is sent next.

Regarding data visualization, the pyqtgraph library was used, since its performance overcomes that of alternatives, such as matplotlib, in applications requiring high graph update rates. The code for graph creation and updating is schematically represented in Figure 7. Besides managing the visualization graphs, the same thread is responsible for storing the data, sending them to the main thread at the end of each monitoring session.

A target data acquisition rate of 10 Hz was initially defined. To maximize system performance and allow this goal to be more reliably met, a compromise was made regarding the graph’s visualization refresh rate. While data can be received at a rate of 10 Hz, refreshing the graphs in the same interval was verified to be too computationally intensive. This way, the graphs are only updated in 0.15 s intervals and show only the last 20 data points. The data acquisition rate was verified through a timer, implemented in the code, activated at the start session signal reception, and stopped at the stop session signal emission.

Pyinstaller was employed to create an executable app for the software which can be installed and run on any Windows 10 computer without the need for any additional file or app. This makes the software easier to distribute and end-user friendly.

### 2.4. Case Study Description

To verify the system’s ability to collect useful data for preventive and programmed injection mold maintenance in a real-world setup, a set of consecutive injection molding runs were carried out in normal operation, and with a simulated fault in the extraction system. The monitoring system was used to register and visualize data from the sensors installed in the injection mold. To carry out the tests, a Tederic D80 injection-molding machine (INAUTOM Automação Lda.-Batalha, Portugal) was used, and low density polyethylene (LDPE) of the Dow^TM^ LDPE 780E grade (DOW Europe GMBH, Horgen, Switzerland), distributed by RESINEX (RESINEX Group, Braga, Portugal) was injected into a cup-shaped part mold. The injection molding process parameters are listed in Table 2.

The 3D CAD models of the injected part, sensors, and their installation placement are shown in Figure 8. As displayed, the force sensor was installed in one of the extraction pins, and the three-dimensional accelerometer and gyroscope module was mounted on the extraction plate to monitor vibrations. Additionally, to monitor mold temperature and cavity pressure, a thermocouple was installed in the mold plate and the pressure sensor was installed in the cavity, directly contacting plastic flow.

The test setup, with the monitoring system and injection molding machine, is represented in Figure 9. A total of 110 cycles were continuously recorded, with the first 54 Injection molding runs in normal operation. In the runs that followed, a fault was simulated in the extraction system by gripping a screw against the pin in which the force sensor had been installed. The fault was simulated multiple times, by retightening the screw.

## 3. Results and Discussion

From the initial tests and characterization performed on the assembled system, it can be verified that the developed software allows the visualization of information relevant to the injection molding process in a simple and intuitive GUI. It also allows data acquisition within the initially targeted 10 Hz rate. Higher acquisition rates may be possible, but, in the current system version, it cannot be guaranteed. Still, in the event of a visualization graphs delay, the acquired data can be stored anyway, with correct timestamps, and be used posteriorly. It must be considered, however, that the acquisition and graph visualization refresh rates are dependent on machine performance and should be verified independently for each new computer.

The real-world context tests showed the system’s suitability for acquiring the data from sensors installed in the molding tool. Data from the injection molding runs, both in normal operation and with simulated failure in the extraction system, can be observed in Figure 10, with no additional data processing. A gray line was used in each subfigure to indicate where the extraction pin fault started to be applied. From cavity pressure data, it is possible to identify the multiple injection molding cycles, with a peak and following dip corresponding to a cycle. The temperature graph dips and long pauses in the pressure graph correspond to the moments at which the screw used to simulate the extraction pin defect was tightened. After these pauses, the highest recorded force sensor values were registered, followed by considerably lower peaks, indicative of the screw carving a path onto the pin, thus progressively providing lower friction. However, even several cycles after fault introduction, the registered force peak values were still higher than under normal operation, on average.

A better direct comparison between normal and faulty operation cycles is given by Figure 11. On it, data from the cavity pressure sensor (Figure 11a), force sensor (Figure 11b), vibration (acceleration in the OX axis, Figure 11c, OY, Figure 11d, and OZ, Figure 11e), can be viewed, with the injection cycles overlapped and identified as blue for normal operation and red for simulated fault operation. Regarding cavity pressure data, it is possible to observe a regular operation cycle that does not align with the remaining ones. This is due to the longer time taken by the operator to start the first run, after the mold was closed. The remaining cycles, however, do align, both in regular and simulated fault operation. Moreover, attesting the system’s capability to acquire useful data in the scope of general injection molding process monitoring, and not only for predictive maintenance, the pressure curves allow a clear identification of the distinct process phases. More specifically, the registered curves are consistent with the general shape presented in [21], being characterized by a steep increase in cavity pressure during the filling phase, after which the maximum pressure is reached. It is then followed by the packing or holding phase during which cavity pressure registers a plateau slightly under peak pressure that is kept until the polymer freezes at the gate. With the gate closed, pressure drops until it reaches a lower plateau, with only residual pressure while the part finishes cooling. At the end of this phase, the mold opens and the part is extracted. The ejection phase can also be easily identified in Figure 11, as it is characterized by the drop in cavity pressure to atmospheric pressure, the peaks in the force values, and the most significant vibrations seen in the acceleration graphs. As observed before in Figure 10, Figure 11 shows clear differences in force amplitude reached in the cycles affected by the simulated fault. Additionally, cycles in which part extraction took longer than normal can be identified and coincide with the extraction pin initially jamming due to the added friction. The data collected from the 3D accelerometer could be used to identify and monitor different machinery movements and respective process phases, similar to the results presented by Moreira et al. [8] and Brunthaler et al. [22]. As broadly verified in those works, the most noticeable vibrations in amplitude, and with a significant duration, were observed for the part ejection phase. Mold closing is identifiable, in the case of this work, at the beginning of the cycles and near the 25 s mark, the latter being more obvious in the OZ acceleration graph. Data displayed after the 25 s mark corresponds to cycles with a longer pause associated with them, or to cycles that required a longer extraction phase due to the introduced fault. Mold opening vibrations can be seen in the graph at 20 s. There are three small disturbances that can be viewed in the acceleration graphs and that are common between the different acceleration directions and multiple cycles. The first may correspond to the screw advancing before the filling phase start, the second to the gate freezing and the third to the screw being retracted. Differences in acceleration components’ amplitudes and time for stabilization can also be observed. This demonstrates the usefulness and viability of the developed monitoring system and similar proposals. The remaining sensors did not show any clear difference between normal operation and otherwise, thus the emphasis on vibrations and force sensors.

From a broader point of view, besides the specific objectives leading to this system’s development having been achieved, its current version suffers from some limitations that need to be addressed and overcome. An example is the fact that, reading the defined sensors is possible, and these sensors cover a reasonable variety of variables that can be monitored; however, the system still has a limited flexibility to easily incorporate other kinds of sensor, with software needing the biggest improvement on that regard. The inclusion of a monitoring mode without data visualization in real-time could be useful as well, allowing data acquisition at faster rates. The capability to directly send the data to a remote database can also be an important future development. Furthermore, there is a need for IoT-enabling the system, as it will allow for wireless data transmission between sensor modules and the computer, increasing its practicality and reducing the risk of damage through wires caught during machine and mold movements. These limitations can be tackled in future works based on this one, mentioned in Section 4.

## 4. Conclusions

The system described in this work has shown data acquisition capabilities compatible with the initially set objectives. Data from up to six sensors installed in the molding tool can be acquired at a rate of 10 Hz, including pressure, temperature, force, and vibration. For applications in which the aim is the simple data acquisition from the mold monitoring sensors, while allowing real-time visualization, in its current form, the system presents a viable alternative to the use of the ones commercialized by the traditional monitoring systems suppliers. Those suppliers generally present more complex solutions, integrating a range of other functionalities, crucial in certain industry applications, making them, however, significantly more expensive. This way, the developed system can be placed as a sensible, reduced cost proposal for mold monitoring in research and/or educational context. Thus, it is fit for the intended application: data collection from sensors installed in the mold for its preventive and programmed maintenance.

The developed software can, in future versions, be adapted in a relatively simple and inexpensive way, showing high potential to become a truly useful tool for advanced mold monitoring algorithms development. A change that can be implemented is the real-time comparison between measured values and an acceptable or ideal working value range. This would be coupled with the development and implementation of an algorithm with an output that allows identifying data patterns close to or over the operational limits, in-process. Thus, the software would be able to perform the early detection of the need for maintenance, as well as suggesting or even autonomously implementing parameter alterations for fault compensation or require process halting. Another adaptation that can improve the system, making it more practical and physically flexible, is enabling wireless communication between the microcontrollers and computer. Monitoring online, mobile, or otherwise, devices, would be a further step in that direction.

## Figures and Tables

**Figure 1 sensors-23-03569-f001:**
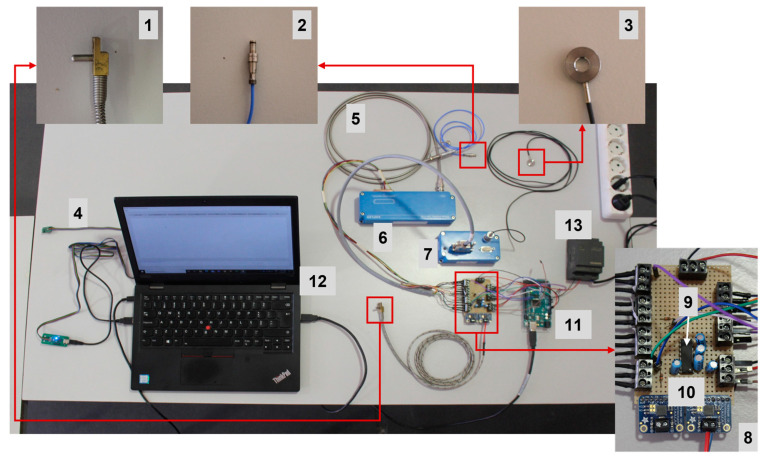
Monitoring system hardware with the components identified according to the numbers in Table 1.

**Figure 2 sensors-23-03569-f002:**
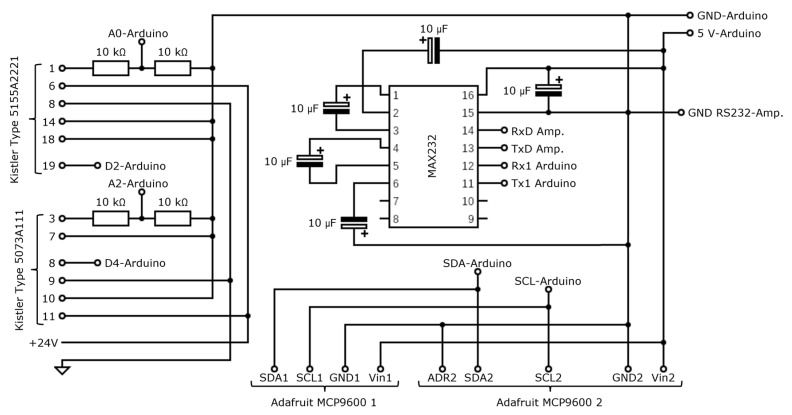
Schematic representation of the connections created for interface between the Arduino, charge amplifiers and voltage amplifier and ADC modules.

**Figure 3 sensors-23-03569-f003:**
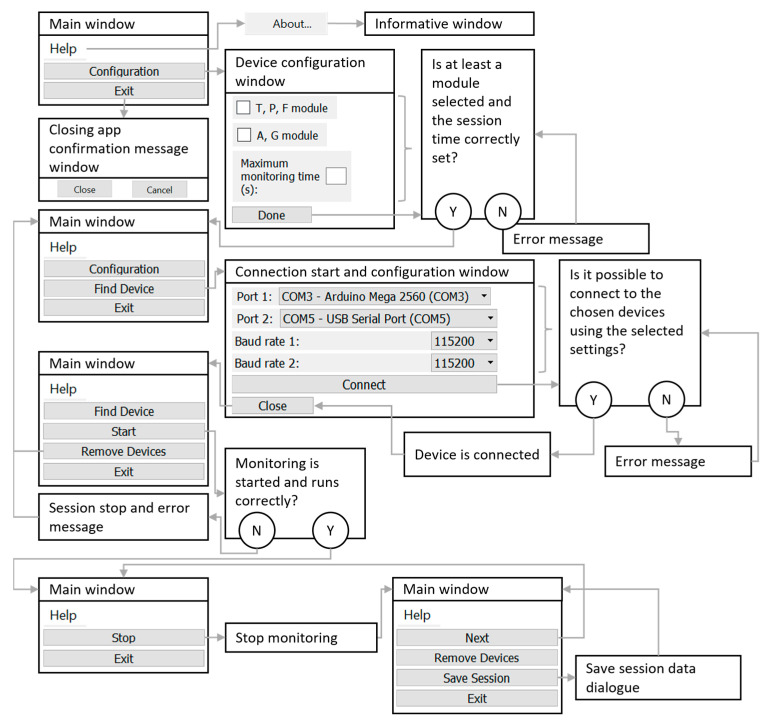
Schematic representation of the GUI, with the available clickable buttons at each operation state and window, as well as the window and operation states that are triggered by clicking each button.

**Figure 4 sensors-23-03569-f004:**
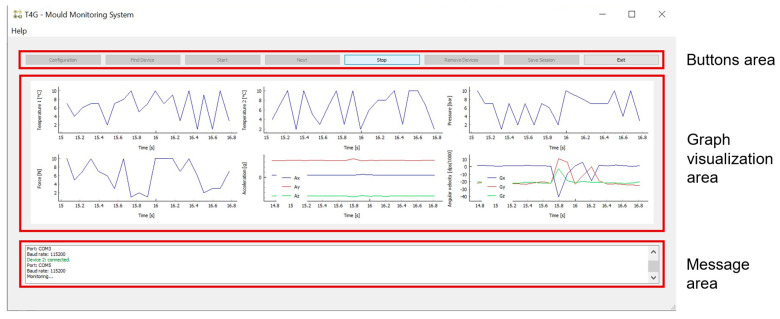
Main window during a monitoring session, with the main zones identified and highlighted.

**Figure 5 sensors-23-03569-f005:**
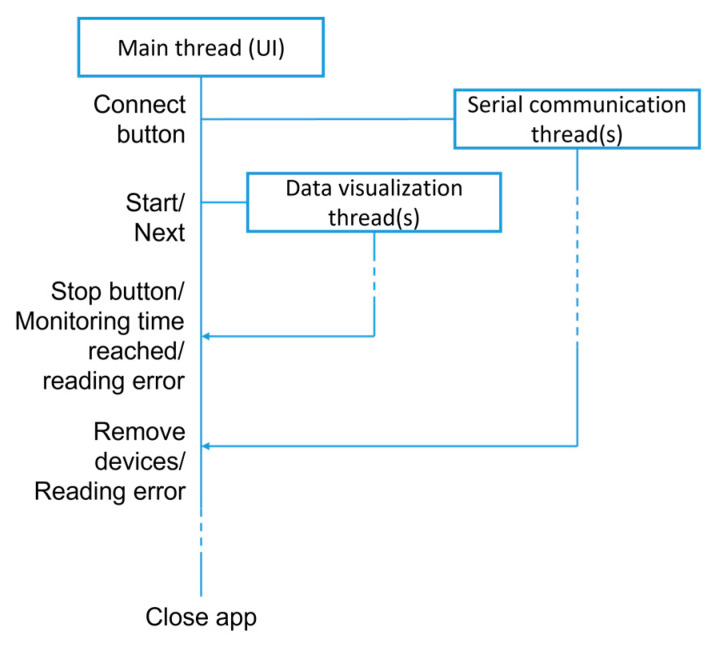
General code structure developed using multithreading.

**Figure 6 sensors-23-03569-f006:**
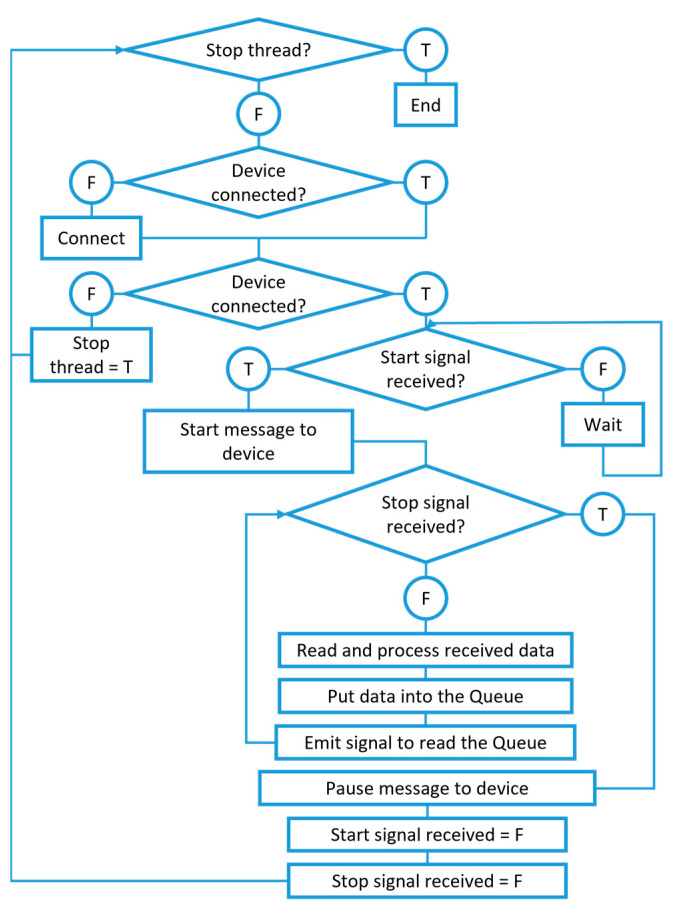
Serial communication thread code structure representation.

**Figure 7 sensors-23-03569-f007:**
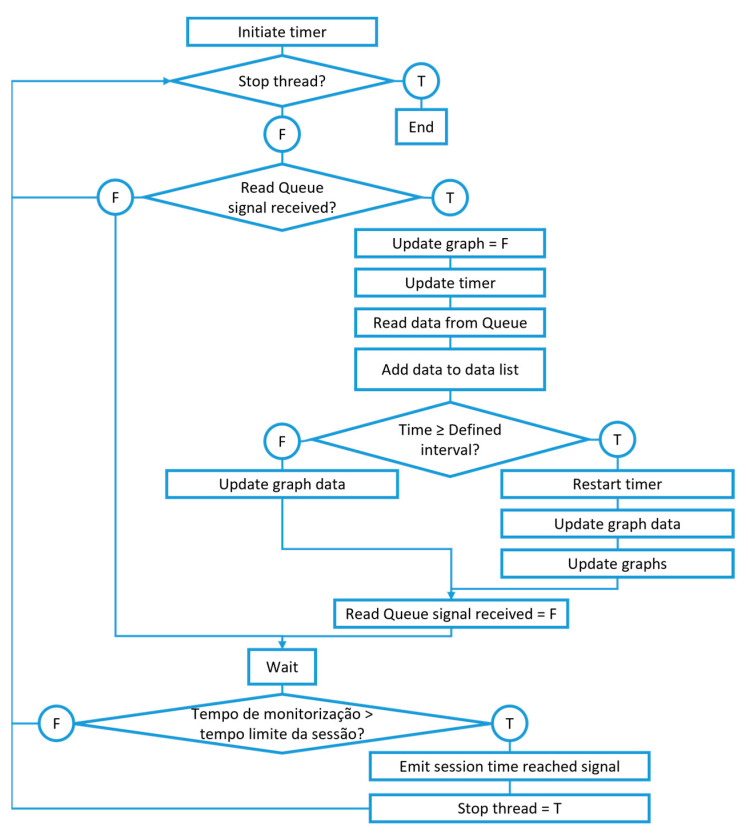
Graph thread code structure schematic.

**Figure 8 sensors-23-03569-f008:**
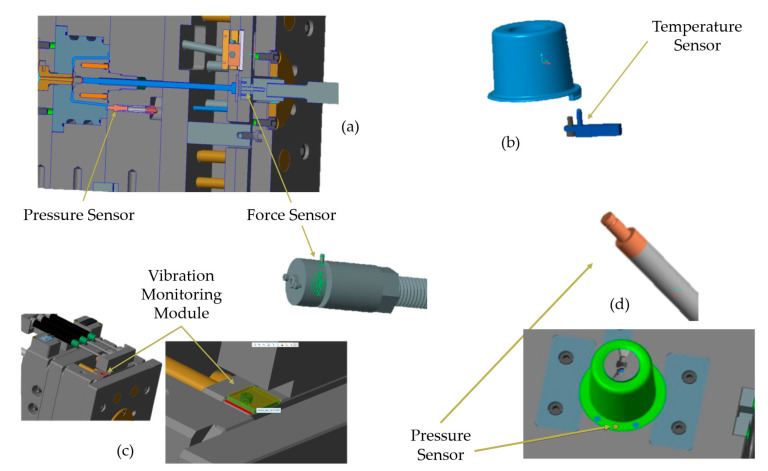
Detailed view of the CAD models representing the injected part, sensors, and their installation positions. (**a**) Section view of the mold with pressure and force sensors placement, and force sensor installation detail; (**b**) Injected part 3D model and relative temperature sensor position; (**c**) Vibration monitoring module position; (**d**) Pressure sensor 3D model and its position relative to the part.

**Figure 9 sensors-23-03569-f009:**
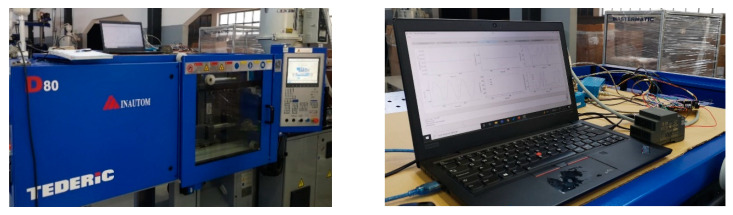
Test setup with the working monitoring system and injection molding machine.

**Figure 10 sensors-23-03569-f010:**
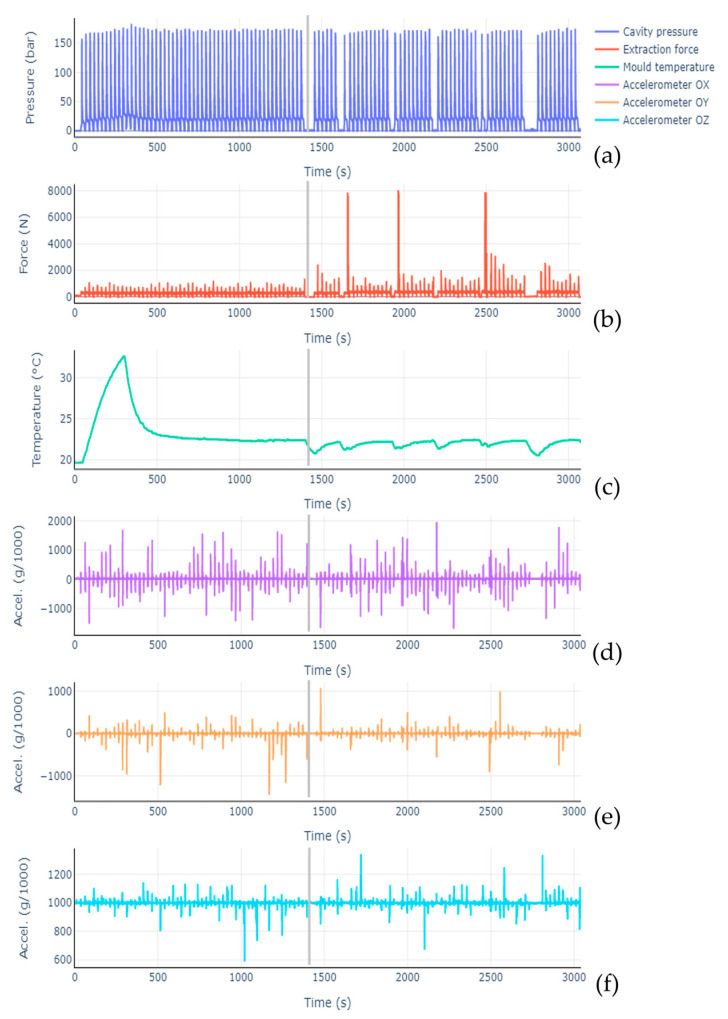
Graphs corresponding to the data collected both under normal operation conditions and with successive simulated faults in one of the extraction pins. (**a**)—Pressure sensor, (**b**)—Force sensor, (**c**)—Temperature sensor, (**d**)—Accelerometer (OX axis), (**e**)—Accelerometer (OY axis), (**f**)—Accelerometer (OZ axis).

**Figure 11 sensors-23-03569-f011:**
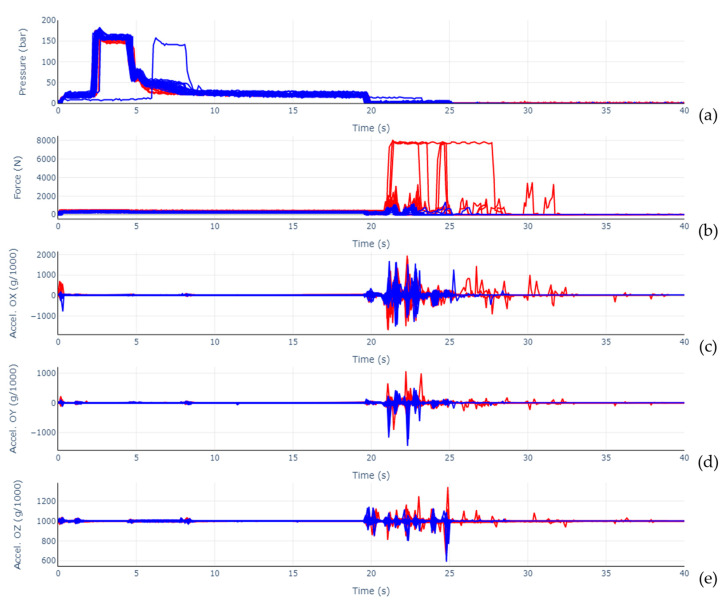
Sensor data for each cycle, overlapped and colored in blue for the regular operation scenario and in red for the simulated fault operation scenario. (**a**)—Pressure sensor, (**b**)—Force sensor, (**c**)—Accelerometer (OX axis), (**d**)—Accelerometer (OY axis), (**e**)—Accelerometer (OZ axis).

**Table 1 sensors-23-03569-t001:** List of the main components used to develop the monitoring system and corresponding designations.

Component	Designation
1—Thermocouple	HASCO Z1295/1 (HASCO Hasenclever GmbH + Co KG, Lüdenscheid, Germany)
2—Pressure sensor	Kistler Type 6157B (Kistler Group, Zurich, Switzerland)
3—Force sensor	Kistler Type 9133B (Kistler Group, Zurich, Switzerland)
4—Vibration monitoring module	Sensors: ST LSM6DSL (STMicroelectronics, Geneva, Switzerland); Microcontroller: Microchip PIC18F27K42 (Microchip Technology Inc., Shanghai, China)
5—Pressure sensor extension cable	Kistler Type 1661A (Kistler Group, Zurich, Switzerland)
6—Pressure sensor charge amplifier	Kistler Type 5155A2221 (Kistler Group, Zurich, Switzerland)
7—Force sensor charge amplifier	Kistler Type 5073A111 (Kistler Group, Zurich, Switzerland)
8—Arduino interface circuit	Made inhouse
9—Device driver chip to communicate with component 7	Texas Instruments MAX232 (Texas Instruments Inc., Dallas, TX, USA)
10—Thermocouple signal amplifier	Adafruit MCP9600 (Adafruit Industries LLC, New York, NY, USA)
11—Arduino	Arduino Mega 2560 R3 (Arduino, Monza, Italy)
12—Computer	Laptop used in tests: Lenovo ThinkPad L380 (Lenovo Group Ltd., Hong Kong, China)
13—Power supply	Generic

**Table 2 sensors-23-03569-t002:** List of the main process parameters used during the test runs.

Parameter	Value
Barrel temperature zones (°C)	Nozzle: 220; Z1: 220; Z2: 210; Z3: 200
Mold temperature (°C)	20
Hydraulic injection pressure (bar)	100
Injection time (s)	0.8
Hydraulic packing pressure (bar)	20
Packing time (s)	4
Cycle time (s)	19

## Data Availability

The raw data collected during the case study can be found at: https://github.com/TEPGomes/OpenMMS-T4G/blob/cfa6e23c7fc02a645e31e06d299021cb0a3ce3e7/Real_World_Test/Case_Study_Raw_Data.csv (accessed on 23 March 2023).

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
