# Peer review of "Development of an Open-Source Injection Mold Monitoring System"

_sensors, 2023, doi:10.3390/s23073569_

Round 1

Reviewer 1 Report

The author describes a system based on open-source software and firmware, which can be used to monitor an injection mold and  to visualize data corresponding to a maximum of six sensors, at a rate of 10 Hz, and recording the data for later usage and analysis.  A software was developed and can receive data from both the Arduino module and a second module. This work is useful for practical injection molding, however, the innovation seems not enough for being published in this journal or from academic point of view.

In Figure 9, the pressure curve seems not similar to conventional pressure curve. What is the reason? Moreover, the figures should be improved. For example, in Figure 4, the text is too small and it is hard to see the text clearly. In Figures 9 and 10, "a, b, c..." should be labelled in the corresponding images. In Figure 10, there is no pressure sensor's result.

Author Response

Dear reviewer

Thank you for your valuable and highly pertinent feedback. The comments, improvement suggestions, and other inputs have been taken into account and were used to enhance the overall quality of the paper. Detailed answers to each question, comment, and suggestion are presented below.

1 - This work is useful for practical injection molding, however, the innovation seems not enough for being published in this journal or from academic point of view.

Answer: The authors understand your point of view, nevertheless, they believe that the approach is novel and useful, not only for this particular application, but also for monitoring other production processes aligned with the industry digitalization. This has been made clearer in the manuscript, where the problem statement and novelty have been further highlighted. (L18 - L21; L165 - L181)

2 - In Figure 9, the pressure curve seems not similar to conventional pressure curve. What is the reason?

Answer: The reason for this is that, in the pressure graph, multiple consecutive injection cycles are represented uninterruptedly, instead of a single one. This way, due to the wide time interval represented, the individual pressure curves of each cycle appear as peaks and dips. This does not allow for the detailed distinction of cycle phases or cycle pressure curve shapes. To allow a closer visualization of the cycle curves, a subfigure with overlayed pressure cycles has been added to Figure 11 (L461), where it was missing, despite being mentioned in the caption. Due to other changes in the manuscript, Figure 9 is now Figure 10.

3 - Moreover, the figures should be improved.

3.1 - For example, in Figure 4, the text is too small and it is hard to see the text clearly.

Answer: The authors acknowledge this limitation in Figure 4. The scale of the Figure has been left as seen in the manuscript since its objective is to show the general appearance of the GUI’s main window and convey the function of each of its main areas. Making it big enough so the text would be readable has been considered, but it was realized that it would not add information relevant enough to justify occupying such a significant amount of relative space in the publication. Meanwhile, Figure 3 (L267) allows the reader to understand what buttons are available and clickable in each window and according to the operation status.

3.2 - In Figures 9 and 10, "a, b, c..." should be labelled in the corresponding images.

Answer: Subfigure letter identifiers have been added in both Figure 9, now Figure 10, (L392) and Figure 10, now Figure 11, (L461) as suggested.

3.3 - In Figure 10, there is no pressure sensor's result.

Answer: Figure 10, now Figure 11, has been updated with the overlapped pressure graphs corresponding to each cycle, now matching the caption. (L461)

Reviewer 2 Report

Dear authors, thank you for the submission. The manuscript is well-written, however, there are some points that must be improved, described below.

Abstract: Please, improve the structure of your abstract. Lots of information about the subject and justification of this study. Consider improving your abstract with more results instead of introduction.

Keywords: All keywords are not a MeshTerm.

L68-69: The development of a promisor low-cost monitoring system cited through an International Conference does not appear to be scientifically appropriate. Consider using a reference from this study, considering a peer-reviewed publication.

L93-95: This sentence seems to be the impact of your study on future studies or application in the application, in addition to bringing a small limitation “despite needing improvement”. Consider changing the position of this sentence to discussion topic.

Table 1: Consider inserting the complete reference of each manufacturer, with city and country, when applicable.

Results and discussion: This topic is named “Results and discussion”, so an appropriate discussion needs to be done, using the scientific literature corroborating or not with your results.

Author Response

Dear reviewer

Thank you for your valuable and highly pertinent feedback. The comments, improvement suggestions, and other inputs have been taken into account and were used to enhance the overall quality of the paper. Detailed answers to each question, comment, and suggestion are presented below.

1 - Abstract: Please, improve the structure of your abstract. Lots of information about the subject and justification of this study. Consider improving your abstract with more results instead of introduction.

Answer: The Abstract has been reworked according to this suggestion. (L15 – L21; L25 – L30)

2 - Keywords: All keywords are not a MeshTerm.

Answer: The authors thank the reviewer for highlighting this, as MeSH terms make the article more likely to be found. MeSH terms have been included in the Keywords. Other terms were left as they were (with the initial letters of each word being changed to upper case, for coherence), some because they are widely used in the field, despite there being no MeSH term for them, and others because, despite not being MeSH terms, they are Entry Terms in the MeSH database for a different term. (L31 – L32)

3 - L68-69: The development of a promisor low-cost monitoring system cited through an International Conference does not appear to be scientifically appropriate. Consider using a reference from this study, considering a peer-reviewed publication.

Answer: The authors do agree that ensuring the scientific validity and relevance of the cited or referenced works is of the utmost importance. Often, works published in international conference proceedings do not go through the peer-review process, as implied by the reviewer. However, the referenced work was published in the 2021 IEEE International Conference on Computational Intelligence and Virtual Environments for Measurement Systems and Applications (CIVEMSA). On its Author Center webpage (https://conferences.ieeeauthorcenter.ieee.org/understand-peer-review/), IEEE states that “IEEE requires all conference papers go through the peer review process before publication.”, which leads us to trust this publication is peer-reviewed and, to our knowledge, has no other corresponding peer-reviewed publication.

4 - L93-95: This sentence seems to be the impact of your study on future studies or application in the application, in addition to bringing a small limitation “despite needing improvement”. Consider changing the position of this sentence to discussion topic.

Answer: To improve the manuscript according to this comment, the original phrase was changed in the Introduction. It was also guaranteed that the information conveyed by it is present in the discussion topic. (L184 – L185; L465 – L467)

5 - Table 1: Consider inserting the complete reference of each manufacturer, with city and country, when applicable.

Answer: The reference of each manufacturer and respective city and country have been added in Table 1, when applicable. (L202 -L221)

6 - Results and discussion: This topic is named “Results and discussion”, so an appropriate discussion needs to be done, using the scientific literature corroborating or not with your results.

Answer: Results and discussion topic has been improved, including results corroboration with the existent literature. (L354 – L481)

Reviewer 3 Report

Dear Authors,

First, identify the right product from the injection molding machine and clearly represent your problem statement.

What are the exact process parameters? How can you control the parameters and prove that clear optimization of each factor with a suitable contribution through standard techniques?

IoT integrated support is exactly missing from your experiment and clearly explains the abnormal prediction of the injection molding machinery and what are the remedial steps to be taken care of for smooth control of the machinery.

Author Response

Dear reviewer

Thank you for your valuable and highly pertinent feedback. The comments, improvement suggestions, and other inputs have been taken into account and were used to enhance the overall quality of the paper. Detailed answers to each question, comment, and suggestion are presented below.

1 - First, identify the right product from the injection molding machine and clearly represent your problem statement.

Answer: Authors wish to thank the reviewer for the suggestion. According to it, the paper now features an additional figure where a model of the case study part, mold, and sensors positions are represented. The used material grade has been specified as well. (L322 – L324; L328 – L329; L342 – L347) Regarding the problem statement, it has been made clearer both in the Abstract and Introduction sections. (L18 - L21; L165 - L181)

2 - What are the exact process parameters?

Answer: To answer this pertinent question, a table - Table 2 - has been added with the main process parameters (L326 – L327). Mention to it in the text has been added as well (L325)

3 - How can you control the parameters and prove that clear optimization of each factor with a suitable contribution through standard techniques?

Answer: While the system presented In this paper does not currently include a corrective action suggestion feature, which can be achieved through the incorporation of fault prediction algorithms in the software, nor a corresponding automatic process parameter compensation or process halting through direct control of the machine, these upgrades are predicted as future works (L502 – L508). Despite this, in its current form, the monitoring system can already be used for early detection of faults through the analysis of the acquired data after each session and, during the process, by experienced operators that, for example, by observing the real-time data for the force sensor, may be able to detect abnormally high values and report them.

4 - IoT integrated support is exactly missing from your experiment and clearly explains the abnormal prediction of the injection molding machinery and what are the remedial steps to be taken care of for smooth control of the machinery.

Answer: Indeed, one of the most immediate improvements that will be made to the system, in the future, is enabling IoT integration. This is mentioned in the manuscript as well (L473 – L477; L508 – L511). The authors do not understand what is meant by “clearly explains the abnormal prediction of the injection molding machinery and what are the remedial steps to be taken care of for smooth control of the machinery”, as the machinery prediction and its smooth control are not features claimed to be currently achieved in the scope of the work reported in the paper.

Author Response

Dear reviewer

Thank you for your valuable and highly pertinent feedback. The comments, improvement suggestions, and other inputs have been taken into account and were used to enhance the overall quality of the paper. Detailed answers to each question, comment, and suggestion are presented below.

1 - Highlight novelty of this work by writing comprehensive problem statement.

Answer: The authors thank the reviewer for his suggestion. It has been followed, and a more comprehensive and clearer problem statement is now part of the manuscript. (L18 - L21; L165 - L181)

2 - The caption of table 1 does not match with contents of the table. Please revise caption.

Answer: The caption has been revised (L201 - L202).

3 - There are several typo/grammatical mistakes in the English. Correction is suggested.

Answer: The manuscript has been searched for typo/grammatical mistakes, and the ones that were identified have been corrected throughout the manuscript.

4 - The authors may like to increase the number of references at least to 20 by incorporating some more updated references.

Answer: The authors appreciate the suggestion, and, in the process of improving the quality of the paper, the number of references has been increased to 22. (L121 – L129; L419; L434)

Round 2

Reviewer 2 Report

Dear authors, the reviews are adequate, so I'm considering this study for publication. Thank you for your kind and objective answers.

Reviewer 3 Report

Dear Authors,

I am suggesting to the authors for implementing the future scope of the work further enhancement.